# Antifouling Performance of Carbon-Based Coatings for Marine Applications: A Systematic Review

**DOI:** 10.3390/antibiotics11081102

**Published:** 2022-08-14

**Authors:** Francisca Sousa-Cardoso, Rita Teixeira-Santos, Filipe J. M. Mergulhão

**Affiliations:** 1LEPABE—Laboratory for Process Engineering, Environment, Biotechnology and Energy, Faculty of Engineering, University of Porto, Rua Dr. Roberto Frias, 4200-465 Porto, Portugal; 2ALiCE—Associate Laboratory in Chemical Engineering, Faculty of Engineering, University of Porto, Rua Dr. Roberto Frias, 4200-465 Porto, Portugal

**Keywords:** carbon nanomaterials, graphene, graphene oxide, carbon nanotubes, antifouling coatings, antibiofilm activity, marine biofouling

## Abstract

Although carbon materials are widely used in surface engineering, particularly graphene (GP) and carbon nanotubes (CNTs), the application of these nanocomposites for the development of antibiofilm marine surfaces is still poorly documented. The aim of this study was, thus, to gather and discuss the relevant literature concerning the antifouling performance of carbon-based coatings against marine micro- and macrofoulers. For this purpose, a PRISMA-oriented systematic review was conducted based on predefined criteria, which resulted in the selection of thirty studies for a qualitative synthesis. In addition, the retrieved publications were subjected to a quality assessment process based on an adapted Methodological Index for Non-Randomized Studies (MINORS) scale. In general, this review demonstrated the promising antifouling performance of these carbon nanomaterials in marine environments. Further, results from the revised studies suggested that functionalized GP- and CNTs-based marine coatings exhibited improved antifouling performance compared to these materials in pristine forms. Thanks to their high self-cleaning and enhanced antimicrobial properties, as well as durability, these functionalized composites showed outstanding results in protecting submerged surfaces from the settlement of fouling organisms in marine settings. Overall, these findings can pave the way for the development of new carbon-engineered surfaces capable of preventing marine biofouling.

## 1. Introduction

Carbon nanomaterials, such as graphene (GP), carbon nanotubes (CNTs), fullerenes, and diamond-like carbon, are recognized for their proven antimicrobial and antiadhesive properties [1]. Graphene consists of a single-layer sheet of sp^2^-hybridized carbon with a two-dimensional honeycomb structure [2]. GP materials show a high specific surface area, electron conductivity, and thermal stability, making them attractive for applications like photocatalysis, energy production, and storage [3,4]. These nanocomposites stand out as some of the strongest and thinnest materials available and have significantly better electrochemical properties than CNTs, which are formed by rolling up either a single graphene sheet, single-walled carbon nanotubes (SWCNTs), or a series of concentric graphene sheets, multi-walled carbon nanotubes (MWCNTs) [5]. CNTs, therefore, exhibit a concentric cylindrical structure with a diameter in the order of nanometers (depending on the number of walls) and a length of several microns (100 µm) extendable to up to a few millimeters (about 4 mm) [6]. Due to their unique properties, such as a remarkable mechanical strength, high thermal conductivity, and structural stability, CNTs are promising nanomaterials for several applications, namely in the industrial, environmental, and medical fields [7,8,9].

The antimicrobial and antifouling (AF) performance of carbon nanomaterials, as well as their outstanding mechanical properties, have led to their application in coatings for marine vessels and underwater structures and devices [10,11,12]. Biofouling is caused by the adhesion of microorganisms to surfaces, which rapidly grow and form a biofilm, building the basis for the further settlement of macroorganisms [13,14]. This biological process has a high impact on several industries, compromising, for instance, the performance of membrane separation processes and water filtration devices and deteriorating marine surfaces [15,16,17,18]. In cooling water circuits from power plants, biofouling leads to efficiency losses and accelerates the deterioration due to biocorrosion by about 20% [19]. In turn, in marine environments, submerged surfaces are rapidly colonized by different types of micro- and macroorganisms. Marine biofouling presents several economic, industrial, environmental, and health-related consequences (Figure 1) [20]. The gradual attachment of fouling organisms to watercraft promotes the increase of the vessel’s drag. It has been shown that the increased roughness caused by a heavily fouled ship hull can result in a power penalty of 86% at cruising speed and even light fouling can generate up to 16% penalty [21]. Consequently, more frequent ship maintenance and higher fuel consumption are required to sustain a given navigation speed, therefore increasing the costs related to that vessel and its circulation [22,23]. In fact, it is estimated that transport delays, hull repairs, and maintenance operations lead to losses of about 150 billion USD per year [24]. Additionally, the requirement for high fuel consumption leads to higher greenhouse gas emissions, negatively impacting air quality and causing severe environmental issues, such as global warming and climate change [25]. Fouled hull vessels also play a crucial role in the introduction of invasive exotic species into non-native environments since attached organisms can be transported over greater distances, impacting the local ecosystem and, ultimately, harming global biodiversity [26]. Furthermore, marine biofouling on marine facilities, such as underwater support structures, sensors, and housings, also poses an important concern [27]. Besides contributing to the deterioration and corrosion of these surfaces, the attachment of fouling species to marine devices used for monitoring can interfere with the measurements performed [28,29].

Several approaches have been used to tackle biofouling. These strategies can prevent, and delay biofilm formation or control already developed biofilms [24,30,31]. While preventive measures comprise both antimicrobial and AF surfaces, control procedures can involve the use of bacteriophages, QS inhibitors, the disruption of the biofilm matrix, as well as the use of matrices with self-cleaning properties [32,33,34,35,36].

During the twentieth century, the development of AF coatings emerged as a new approach to protecting ship hulls from biofouling. Among all the coatings developed and tested, tributyltin (TBT) stands out as one of the most effective in biofouling mitigation [37]. However, TBT was exclusively used in the 1970s, in combination with several copper compounds, since it was discovered that TBT-based compounds were highly toxic for nontarget organisms. Moreover, they were also able to persist in the marine environment for a longer period than initially estimated [38,39].

The prohibition of organotin compounds such as TBT strongly motivated AF coating manufacturers to find additional solutions that allow a similar biofouling mitigation performance obtained with TBT-based compounds but without adverse environmental impacts. The current challenge is to develop environmentally friendly marine AF coatings that are effective toward a broad range of marine organisms with no negative impact on nontarget organisms. Furthermore, leached coating compounds should have a high degradation rate in the marine environment, as well as a low bioaccumulation rate [38].

Modifications of surface charge potential, hydrophobicity, and surface topography can constitute promising strategies [40]. In this context, the use of carbon nanomaterials as fillers for polymer composites is becoming increasingly relevant [41].

The application of these nanomaterials can increase the mechanical strength of the final composite [42] and its ability to prevent or delay biofouling [43].

Both GP and CNTs have shown good antimicrobial activity against Gram-positive and Gram-negative bacteria, as well as bacterial spores [1,5]. Regarding CNTs specifically, SWCNTs have exhibited significantly higher antibacterial activity than MWCNTs against Gram-positive bacteria [44]. However, the mechanisms of action behind these carbon-based nanomaterials are still not fully understood, due to their complexity and the wide array of factors that may influence their antibacterial activity, including their composition and geometry, as well as the type, morphology, and growth state of bacteria (planktonic or sessile) [7,45]. It is hypothesized that the antibacterial properties of these nanomaterials rely mainly on mechanical factors: their sharp structures act as ‘‘nano-darts’’ that pierce bacterial membranes, leading to cell death [44]. Nevertheless, other authors defend that CNTs produce not only mechanical damage with consequent cell disruption and release of intracellular content (a primary killing mechanism), but also generate oxidative stress [46] (Figure 2). It has also been reported that the length, diameter, surface area, concentration, and chemical modifications of CNTs play a significant role in both the AF and antimicrobial activity of these carbon materials [46,47,48].

As for graphene-based nanomaterials, it is also assumed that both physical and chemical factors come into play. Similar to CNTs, the sharp edges of GP sheets cause membrane damage [49]. Thanks to the large surface area of graphene-based materials, they can also lead to bacterial cell entrapment [50]. Moreover, GP is assumed to be able to induce oxidative stress through the formation of reactive oxygen species (ROS), which disrupt microorganisms’ DNA and proteins (Figure 2). According to the literature, the mechanisms through which GP-based nanomaterials induce cell inhibition/death are not only dependent on their own physical and chemical properties (e.g., dimensions, number of layers, functionalization), but also on factors related to the production of the surface (e.g., graphene loading, nanoparticle dispersion, aggregation) [51,52,53].

Thanks to these appealing properties, the use of these nanomaterials for improved marine AF coatings is currently on the rise [41]. As such, assessing the effectiveness of GP- and CNTs-based coatings in preventing marine biofouling, namely on pioneer bacterial attachment and biofilm formation, can contribute to optimizing and reaching a better understanding of their AF properties. However, the currently available data regarding the potential of carbon-based coatings to prevent marine biofouling need to be critically discussed to assist researchers in the design of improved marine surfaces.

## 2. Results and Discussion

### 2.1. Study Selection and Characterization

A total of 152 articles were found using the PRISMA (Preferred Reporting Items for Systematic reviews and Meta-Analysis) search methodology. This number was increased to 163 studies with the inclusion of 11 records retrieved through other sources (prior searches and references of the chosen publications). After screening out duplicates, a total of 159 articles were assessed based on title and abstract. Out of these, 127 records were disqualified for not meeting the prerequisites for inclusion. Lastly, 2 records were ruled out upon thorough analysis of the remaining 32 full-text publications since these were nonoriginal articles. Therefore, 30 studies were included in the qualitative synthesis (Figure 3).

With the increasing search for environmentally friendly AF materials, the study of carbon nanomaterials and their unique properties has been rising during the past decade. These nanomaterials are generally incorporated into commercial AF paints based on polymeric matrices, such as polydimethylsiloxane (PDMS), to assess how they influence the physicochemical properties and AF performance of coatings.

Due to the appealing physical and chemical properties it possesses, such as high hydrophobicity, low surface energy, adhesion, and endurance, PDMS stands out as one of the most widely used polymers for AF purposes [54]. However, PDMS also presents certain limitations, namely, its mechanical weakness and propensity to become damaged when immersed in a marine environment [55]. Thus, any additive capable of improving the efficacy and durability of PDMS-based AF coatings is extremely promising.

CNTs-modified PDMS composites have gained worldwide attention due to their facile fabrication, ecological stability, and remarkable AF performance [56]. Furthermore, the incorporation of GP or CNTs in polymeric coatings has been shown to improve their physical properties, namely in terms of mechanical strength [42,57], increasing their capability to delay marine biofouling.

In this systematic review, 30 full-text articles focused on carbon materials incorporated into polymeric coatings and their AF properties in the marine biofouling context were reviewed and summarized, as shown in Table 1 and Table 2. Among the selected publications, 12 focused on the incorporation of pristine or modified MWCNTs and 11 used graphene oxide (GO), either on its own or functionalized with other nanoparticles. In addition, five studies addressed the AF performance of other GP forms. PDMS stood out as the most used nanocoated polymer, with a total of 14 out of the 30 studies.

Based on the reviewed articles, Figure 4 represents the evolution that the number of studies focused on carbon-nanocoated polymeric materials for marine AF purposes has shown over the years.

Considering that no restrictions were applied to the search in terms of publication year, the fact that the first article identified on the subject dates to only 2008 proves that this strategy is fairly recent. In addition, the increasing number of studies published over the last few years confirms the relevance of this topic as a research matter.

In this systematic review, particular attention was given to carbon-based coatings with application in the marine field and their antifouling potential against micro- and macrofoulers.

### 2.2. Graphene-Based Coatings

Several authors have demonstrated the promising AF activity of GP coatings against marine bacteria [58,59] (Table 1). According to Jin, Zhang, et al., under dynamic conditions, graphene-based membranes were able to reduce bacteria adhesion by 40% [59].

However, a records analysis reveals that the current trend is to study the potential of modified/functionalized GP. The functionalization of GP with silver nanoparticles has demonstrated remarkable antibiofilm effects, with a 99.6% inhibition rate for *Halomonas pacifica* and over 80% for *Dunaliella tertiolecta* and *Isochrysis* sp. [60]. Recently, guanidine functionalized GP has also shown promising antibacterial and diatom antiadhesion properties, with reduction rates of up to 95% and up to 99.2%, respectively. Moreover, the field trial revealed no fouling or surface deterioration for 2 months [61].

Furthermore, laser-induced GP coatings reduced *Cobetia marina* surface coverage by up to 80% after 36 h of exposure [62].

These results indicated that functionalized GP coatings can be successfully applied for the development of AF marine surfaces.

### 2.3. Graphene Oxide-Based Coatings

Graphene oxide, the most studied form of GP within the marine AF coatings context, has shown both *in vitro* and *in situ* high AF activity [63,64,65] (Table 1). In fact, surfaces containing GO 0.36 wt% when incubated for 10 days under dynamic conditions almost completely inhibited diatom adhesion [64] (Figure 5).

However, GO is often functionalized with metal nanoparticles, such as silver or alumina, and other compounds. These nanocomposites aim to provide GO with enhanced antimicrobial properties by improving particle dispersibility and strengthening the contact between the carbon nanomaterials and surrounding microorganisms. GO–silver nanoparticles coatings developed by Liu et al. and Zhang and Mikkelsen showed improved antibacterial and antialgal properties, and more than 80% average biofilm inhibition against *H. pacifica* (Figure 6), respectively [66,67]. Besides silver, alumina and silica nanoparticles have also been used in conjunction with GO, having both demonstrated excellent antimicrobial properties against a wide range of organisms [68,69]. In turn, the functionalization of GO with polyaniline/p-phenylenediamine conferred anticorrosion and AF properties to commercialized epoxy coatings [72]. Likewise, the modifications of GO materials with compounds such as cuprous oxide, acrylic acid, or boehmite nanorods, produced AF coatings with high self-cleaning performance and durability in marine environments (up to 6 months) (Figure 7) [70,71,73].

In general, these results suggest that GO composites have promising AF and anticorrosion properties, which are so desirable in the marine industry.

### 2.4. Carbon-Nanotubes-Based Coatings

Up to date, several studies have reported the effectiveness of CNT-based coatings in the prevention and control of marine biofouling. Table 2 describes the studies demonstrating the efficacy of these coatings in marine environments, which refer essentially to MWCNTs.

Concerning the application of pristine MWCNTs (p-MWCNTs) for marine coatings, the obtained results differ. While some studies demonstrated that the incorporation of these carbon nanomaterials on PDMS improves its AF performance by reducing the abundance of pioneer eukaryotic microbes [79] and the adhesion strength of adult barnacles [75,78], other studies revealed that p-MWCNT-based coatings did not affect the settlement of micro- and macrofoulers [76,77].

In addition, recent studies have strived to assess the influence of carboxyl- and hydroxyl-modified MWCNTs on the AF performance of marine coatings. Sun and Zhang performed a thorough study of these modified surfaces by carrying out a two-month field trial focused on the addition of MWCNTs with varying hydroxyl content % (*w*/*w*), carboxyl content % (*w*/*w*), diameter, and length into PDMS coatings. Results showed that the type of MWCNTs had an impact on the coating’s AF behavior, as well as on pioneer eukaryotic communities [55]. Conversely, Ji et al. demonstrated that most carboxyl- and hydroxyl-modified coatings had weak modulating effects on pioneer biofilm communities [81], while Sun and Zhang showed that the AF behavior of these modified CNTs varied for different pioneer biofilm bacteria [55].

In turn, the production of fluorinated MWCNTs polymer-based coatings showed a promising antiadhesion effect against pseudobarnacles [84] and an impressive 98% reduction rate against *E. coli* [85].

Furthermore, the infusion of lubricants, such as silicone oil, into the polymeric matrices was revealed to be a promising approach to MWCNTs-based coatings [74,80,83]. This strategy aims to develop long-term superhydrophobic fouling release (FR) surfaces that leach lubricant over time, creating an isolation oil layer that protects the surface from deformation or damage caused by friction and reinforces its AF properties [83].

Altogether, these data provide important findings that should be considered in the development of new CNTs-based antifouling marine coatings.

### 2.5. Other Carbon-Nanomaterials-Based Coatings

Apart from GP and CNTs, the AF potential of other carbon nanomaterials has been explored. Recently, Luo et al. synthesized atomic chromium–graphitic carbon nitride coatings and demonstrated their *in situ* activity to control marine biofouling for approximately 2 months [86].

### 2.6. Qualitative Assessment

In order to assess the validity of the obtained results and their predictive value, the 30 selected articles were scored according to an adapted MINORS scale (Table 3). Out of a maximum score of 24, the studies obtained a mean score of 21.5 ± 2.0.

All articles clearly stated the aim of the work, presented adequate methodologies, and provided enough information about the composition and fabrication method of the tested coating (criteria one, two, and four; mean score of 2.00). Additionally, 29 out of the 30 articles reported at least three replicates/independent experiments for each assay, as well as an adequate control group (criteria three and five; mean score of 1.93). Moreover, most studies provided sufficient information about the experimental setup used (criterion eight; mean score of 1.90) and implemented appropriate surface characterization methods (criterion six; mean score of 1.87).

Although these results are overall positive, some limitations were also found. It is noteworthy that 17 out of the 30 selected studies were carried out solely *in vitro*, most under static conditions (criterion seven; mean score of 1.73). This is quite unfortunate since experimental setups that mimic the marine environment (e.g., hydrodynamic conditions, day-to-night light variation) or *in situ* studies can be much more reliable for most applications. Moreover, a considerable number of studies did not report the number of organisms (e.g., cell concentration) that the surfaces were exposed to (criterion nine; mean score of 1.76), and only evaluated the performance of AF coatings for short-term adhesion (criterion 10; mean score of 1.76). Lastly, only 12 out of the 30 assessed publications clearly mentioned the implementation of statistical tests appropriate to the dataset (criterion 12; mean score of 0.87). Since a statistical analysis is a crucial part of producing trustworthy results and predictions, this is considered to be a critical flaw of most articles found.

Despite the overall high score of the selected studies, due to the wide array of methodologies used to test the AF properties of the coatings and the lack of proper statistical analysis, it is essential to highlight the importance of conducting further tests with robust, established methodologies and adequately analyzing the results, to facilitate the comparison between studies and draw more reliable conclusions about AF marine coatings. Moreover, one of the key issues in the development of new AF surfaces is the necessary screening of candidate surfaces (usually tested *in vitro* in a first step prior to *in situ* testing) before scale-up and final performance evaluation. Since it has been shown that hydrodynamics can severely affect biofilm formation [87,88,89,90] and gene expression by fouling organisms [91,92], it is recommended that these tests be performed in controlled hydrodynamic conditions that mimic the final application scenario [93].

## 3. Methods

### 3.1. Search Strategy, Study Eligibility, and Data Extraction

Previously published studies evaluating the effectiveness of carbon nanomaterials, namely GP and/or CNTs used to produce AF or FR coatings for the prevention or control of the attachment of micro- or macrofoulers in marine settings were systematically reviewed based on the PRISMA (Preferred Reporting Items for Systematic reviews and Meta-Analyses) Statement [94].

The search was conducted until June 6th, 2022, using two electronic databases, ScienceDirect and PubMed, through selective combinations of relevant keywords: “marine biofilm”, “biofouling”, “antifouling”, “coatings”, “graphene”, and “carbon nanotubes”. Published full-text articles in English were assessed for eligibility. Studies were screened based on title, abstract, and, ultimately, full content.

The reference sections of all included articles were carefully examined for additional articles that were not identified through the database search. The main inclusion criteria were: (1) *in situ* studies concerning the application of GP and/or CNTs composites as AF/FR coatings in the marine environment; (2) *in vitro* studies focused on assessing the properties of GP and/or CNTs composites, including their antiadhesion, antimicrobial, anticorrosion activities, and AF/FR performance against marine micro- and macrofoulers. As for the main exclusion criterion, any nonoriginal articles, such as reviews or reports, were discarded.

For each selected article, information concerning the materials and composition of the coatings used, identification of the tested fouling organisms, experimental setup, and relevant conclusions were extracted and analyzed by two independent reviewers.

### 3.2. Quality Assessment

Selected studies were subjected to a quality assessment procedure based on an adapted Methodological Index for Non-Randomized Studies (MINORS) scale [95]. MINORS is a validated instrument designed to evaluate methodological quality, as well as to detect potential biases in nonrandomized surgical studies. Although there are no methodological indices to measure the risk or the quality of nonclinical studies, the MINORS scale can be adjusted to other scientific contexts and still serve as a valuable quality assessment tool [96]. As such, the MINORS scale was adapted to the specific context of this systematic review and used to evaluate the overall quality and predictive value of each publication.

For each article, 12 parameters were evaluated: (1) clearly stated aim; (2) adequate methodology; (3) detection of bias; (4) proper identification and description of the studied coating; (5) adequate control group; (6) surface characterization; (7) type of study; (8) experimental setup; (9) organisms studied; (10) biofilm formation/fouling assay duration; (11) clarity of the results, and (12) adequate statistical analysis. Each parameter was scored according to a 3-point scale: 0 (not reported), 1 (inadequately reported), or 2 (adequately reported). Therefore, the maximum score a certain publication could achieve was 24.

## 4. Conclusions

Among the solutions found to control marine biofouling, the application of coatings with antifouling properties on submerged surfaces has been the most promising approach. Due to their outstanding properties, carbon nanomaterials have proven to be promising in the production of antimicrobial and antifouling surface coatings.

This systematic review demonstrated that, over the last few years, there has been an increasing interest in the synthesis and evaluation of carbon-based coatings, in particular those containing GP or CNTs, to prevent and control marine biofouling. Most of the reviewed studies investigated the efficacy of modified/functionalized GP or CNTs, which demonstrated improved AF performance compared to their pristine forms.

Although these studies provided promising results, most AF coatings were only evaluated *in vitro* and, therefore, *in situ* or case studies are missing to validate their real-world application. It is also noteworthy that 26.7% of studies only qualitatively evaluated the AF performance of developed coatings or did not disclose the extent of attachment reduction and/or foulers inactivation. Moreover, the degradation or bioaccumulation rates of carbon-based coatings in the marine environment, as well as their toxicity for nontarget organisms were not addressed in the reviewed studies.

The development of more accurate and reliable *in vitro* test methods that are able to mimic the hydrodynamic conditions observed for each target application is of paramount importance in obtaining reliable data. This is crucial because *in situ* tests should not be performed with surfaces that release compounds for which the leaching and toxicity profiles have not been determined. With an increasing environmental conscience from the public and regulatory organizations, novel coatings must be nontoxic to nontarget organisms, durable, and amenable to large-scale production in a sustainable way. This is a multidisciplinary endeavor requiring the involvement of different stakeholders who have to find a common language to address these issues. In a globalized world depending on maritime transportation, the development of more effective AF coatings is of extreme importance.

Overall, this review illustrated the antifouling potential of carbon-based surfaces for marine applications and emphasized the need for further research on this topic.

## Figures and Tables

**Figure 1 antibiotics-11-01102-f001:**
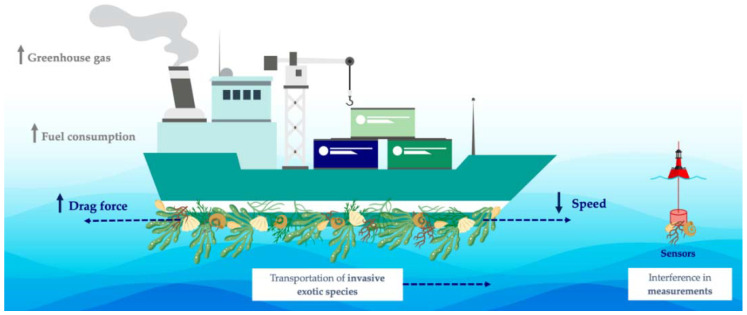
Schematic representation of the negative impact of marine biofouling.

**Figure 2 antibiotics-11-01102-f002:**
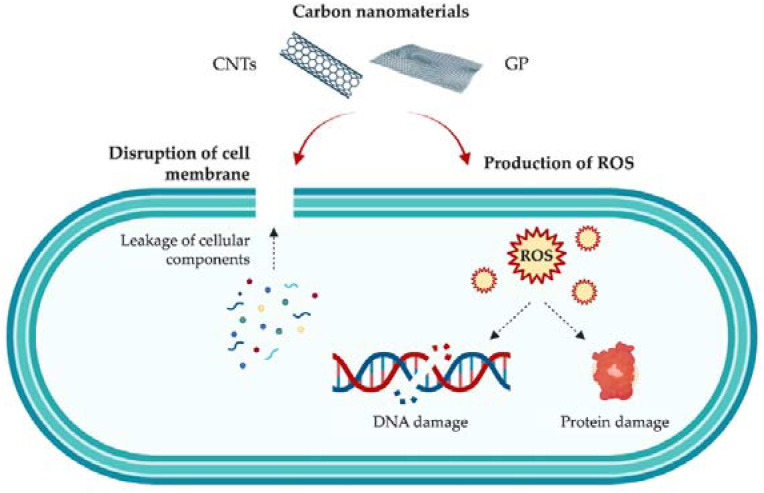
Schematic representation of the mechanisms of action of carbon nanomaterials, namely GP and CNTs, against bacteria.

**Figure 3 antibiotics-11-01102-f003:**
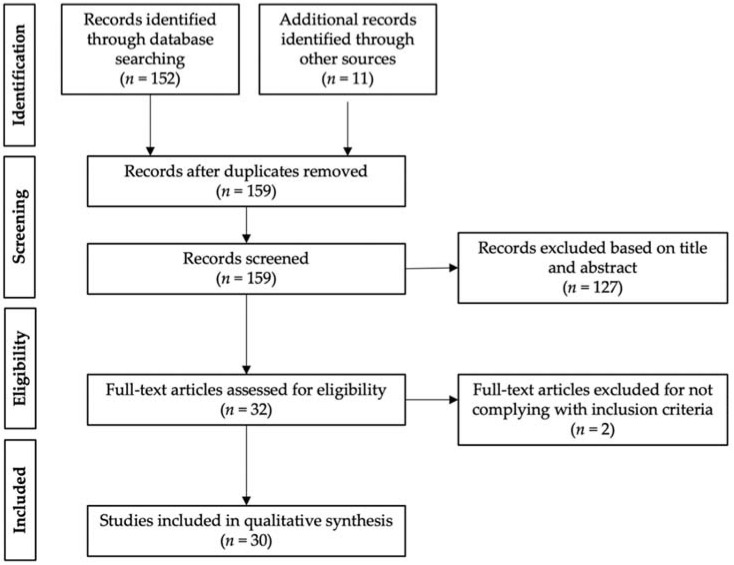
Schematic summary of the PRISMA literature search.

**Figure 4 antibiotics-11-01102-f004:**
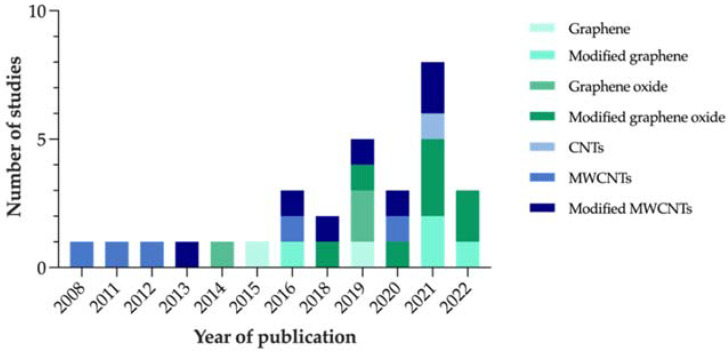
Number of studies on carbon-modified AF marine coatings included in the systematic review, per year of publication.

**Figure 5 antibiotics-11-01102-f005:**
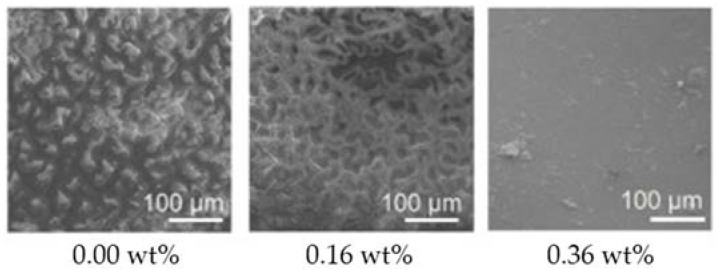
Scanning electron microscopy (SEM) images of diatom adhesion on silicone surfaces containing different GO loadings, after 10 days of incubation under dynamic conditions. Reprinted with adaptations from [64], under the terms of the Creative Commons Attribution (CC BY) license.

**Figure 6 antibiotics-11-01102-f006:**
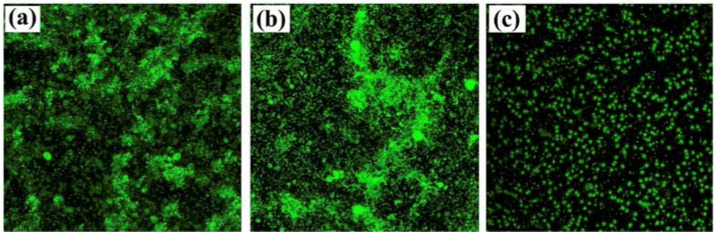
Confocal laser scanning microscopy (CLSM) images of bare polypropylene (PP) (**a**), graphene oxide coated PP (**b**) and graphene oxide/silver nanoparticles coated PP (**c**). Reprinted with adaptations from [67], under the terms of Creative Commons CC BY license.

**Figure 7 antibiotics-11-01102-f007:**
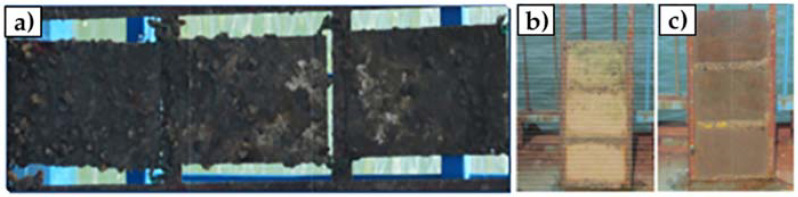
*In situ* marine fouling tests over a period in a sheltered bay connected to the south China sea. Bared panels (90 days) (**a**); Cu_2_O paint-coated surfaces (365 days) (**b**); and GO/Cu_2_O paint-coated surfaces (365 days) (**c**). Reprinted with adaptations from [70], under the terms of the Creative Commons Attribution International License (CC BY 4.0).

**Table 1 antibiotics-11-01102-t001:** Studies focused on graphene-based AF coatings in marine environments.

Coating	Material/Matrix	Organism	Experimental Setup	Main Conclusions	Ref.
Graphene	Silica	*Halomonas* spp.	*In vitro* study Saline solution (0.5 wt%) 20 °C, 72 h	Graphene coatings were effective in decreasing the adhesion and expression levels of adhesion genes of biofilm-producing bacteria *Halomonas* spp.	[58]
	Silicone rubber	*Paracoccus pantotrophus*	*In vitro* study Artificial seawater Quasi-static assay (7 days) Dynamic assay (7 days, varying speeds within the 0.2–0.5 m/s range)	Under quasi-static conditions, the graphene–silicone membranes showed similar AF performance to that of the control surface (rigid polystyrene sheet). Under dynamic conditions, the graphene-based membranes showed better AF performance than the control surface, with around 40% reduction in colony-forming units (CFUs).	[59]
Graphene– silver nanocomposites	Silicon	*Halomonas pacifica*	*In vitro* study Static assay Marine broth 26 °C, 24 h	The nanocomposite displayed significant bacterial biofilm inhibition (99.6% reduction) and antiproliferative effects on marine microalgae (growth inhibition greater than 80%), whereas surfaces coated with graphene alone did not display any AF properties when compared to the control surface.	[60]
		*Dunaliella tertiolecta* *Isochrysis* sp.	*In vitro* study Provasoli medium 4 days		
Guanidine- functionalized graphene	Boron acrylate polymer	*Escherichia coli* *Staphylococcus aureus*	*In vitro* study Luria–Bertani medium 37 °C, 12 h	The coatings showed excellent antibacterial properties (up to 95% reduction) and diatom antiadhesion rates (up to 99%). The field trial revealed no fouling adhesion or surface deterioration.	[61]
		*Phaeodactylum tricornutum* *Nitzschia closterium f. minutíssima* *Halamphora* sp.	*In vitro* study F/2 medium 21 °C, 14 days		
		Marine micro- and macrofoulers	*In situ* study Natural seawater (Yellow Sea, China) 2 months		
Laser- induced graphene	Poly(ether)sulfone	*Cobetia marina*	*In vitro* study Dynamic assay (65 rpm) Artificial seawater 1 and 36 h	Compared with negative control surfaces, laser-induced graphene coatings showed greater initial bacterial attachment (1 h) but up to 80% less bacterial coverage after 36 h. Initial attachment rates were reduced by the application of negative or positive potential.	[62]
Graphene Oxide	Alkyd resin	*Escherichia coli* *Staphylococcus aureus* *Pseudomonas aeruginosa*	*In vitro* study Nutrient medium Room temperature 24 and 48 h	Graphene oxide-coated surfaces greatly reduced bacterial growth (up to 94% loss of cell viability) *in vitro* and biofouling *in situ*.	[63]
		Marine micro- and macrofoulers	*In situ* study Natural seawater (Jeju Sea, South Korea) 3 weeks		
	Silicone rubber	*Triceratium* sp.	*In vitro* study Algal broth medium 25 °C Static assay (8 days) Dynamic assay (10 days, 3.4 m/s linear velocity near the specimens)	Under static conditions, only one of the tested graphene oxide loadings (0.16 wt%) showed slight diatom antiadhesion effects (approximately 12% OD reduction). In the dynamic assay, only one graphene oxide loading (0.36 wt%) showed diatom antiadhesion properties (approximately 67% OD reduction).	[64]
	Polymeric membrane calcium ion selective electrode	Marine bacteria	*In vitro* study Luria–Bertani medium Room temperature 1 h and 5 h	Compared to the noncoated sensor, the proposed graphene oxide-coated sensor displayed significantly improved antiadhesion and bacterial inactivation properties, thus inhibiting the formation of biofilms (around 45% reduction in CFUs).	[65]
Graphene oxide–silver nanoparticles	PDMS-silica	*Escherichia coli*	*In vitro* study Shaking flask method Saline solution (0.9 wt%) 37 °C, 24 h	The coating containing silver nanoparticles showed improved antibacterial (60% greater inactivation rate) and antialgal (up to 17% reduction in surface coverage) properties, in comparison to pristine graphene oxide.	[66]
		*Phaeodactylum tricornutum* *Navicula torguatum* *Chlorella* sp.	*In vitro* study Artificial seawater 24 h		
	Polypropylene	*Halomonas pacifica*	*In vitro* study Static assay Marine broth 26 °C, 24 h	Graphene oxide showed almost no AF properties, while graphene oxide/silver nanocomposites showed more than 80% of biofilm inhibition, as well as no visible fouling by microalgae.	[67]
		Marine microalgae	*In vitro* study Adam medium *^a^* 1 week		
Graphene oxide–alumina nanorods	PDMS	*Micrococcus* sp. *Pseudomonas putida* *Aspergillus niger*	*In vitro* study Nutrient-infused medium 35 °C, 28 days	The nanocomposite showed high adhesion resistance for the selected microorganisms (up to around 95% reduction in the number of bacterial cells).	[68]
		Marine micro- and macrofoulers	*In situ* study Natural seawater 23–28 °C, 3 months	The field trial revealed no fouling or surface deterioration for the nanofilled sample, as opposed to pristine PDMS.	
Graphene oxide–silica nanoparticles	PDMS	*Pseudomonas* sp. *Bacillus* sp.	*In vitro* study Nutrient broth 72 h	The coated surfaces showed up to a 4-Log reduction in total viable cells. Analysis of biofilm architecture confirmed a significant reduction of biomass and biofilm thickness on coated surfaces.	[69]
		Freshwater bacterial culture			
Graphene oxide–cuprous oxide nanoparticles	Acrylic resin	Marine micro and macrofoulers	*In situ* study (0.2–2.0 m below the surface) Natural seawater (South China Sea) Weak water currents (less than 2 m.s^−1^) 90 and 365 days	Bare panels showed an abundant growth of marine organisms within 90 days, while coated surfaces were hardly fouled by marine organisms after 365 days.	[70]
Acrylic acid-modified graphene oxide	Acrylic resin	Marine micro and macrofoulers	*In situ* study Natural seawater (Zhoushan Sea, China) 6 months	Composite-based paint showed great self-polishing AF performance in natural seawater.	[71]
Polyaniline/*p*-phenylenediamine-functionalized graphene oxide	Epoxy resin	Organisms in a simulated marine environment (e.g., guppy fish, *spirulina* algae, and dwarf hair grass)	*In vitro* study Simulated marine environment 25–27 °C, 3 months	The anticorrosion and AF properties of commercialized epoxy coatings were improved by the addition of the functionalized graphene oxide composite.	[72]
Reduced graphene oxide	PDMS	*Staphylococcus aureus* *Kocuria rhizophila* *Pseudomonas fluorescens* *Pseudomonas aeruginosa*	*In vitro* study Nutrient-infused medium 25 °C, 28 days	In laboratory assays, boehmite nanorod composite coating showed higher antimicrobial activity (endurability percentages for Gram-positive, Gram-negative, and fungi of 86.4%, 97.9%, and 85.9%, respectively) in comparison with bare PDMS and reduced graphene oxide/PDMS. The higher self-cleaning and FR performance of the boehmite nanorod composite coating was confirmed by the field trial.	[73]
Graphene oxide–boehmite nanorods		*Candida albicans* *Aspergillus brasiliensis*			
		Marine micro- and macrofoulers	*In situ* study Natural seawater 23–28 °C, 45 days		

Abbreviations: AF—antifouling; CFUs—colony-forming units; FR—fouling release; OD—optical density; PDMS—polydimethylsiloxane. *^a^*, an artificial freshwater.

**Table 2 antibiotics-11-01102-t002:** Studies focused on CNT-based AF coatings in marine environments.

Coating	Material/Matrix	Organism	Experimental Setup	Main Conclusions	Ref.
CNTs	Silicone oil-infused epoxy resin	*Chlorella* sp. *Phaeodactylum tricornutum*	*In vitro* study Artificial seawater 22 °C, 21 days	CNTs/epoxy resin coating showed substantially lower algae settlement than bare epoxy resin. Silicone oil-infused CNTs/epoxy resin coating showed even greater inhibition of algae biofilm formation (up to 90% cell reduction).	[74]
MWCNTs	PDMS	*Ulva linza* *Balanus amphitrite*	*In vitro* study Dynamic assay (similar to the hull of a ship travelling at 15 knots) Artificial seawater 28 °C 45 min for the settlement of zoospores; 7 days for sporeling growth 24 h for initial settlement; 3 months for adhesion strength determination	The release of sporelings was improved by the addition of MWCNTs (approximately 60% of sporeling removal). A significant reduction in adhesion strength of adult barnacles growing on MWCNTs/PDMS was observed.	[75]
		*Ulva linza*	*In vitro* study Dynamic assay Artificial seawater 18 °C, 6 days	The incorporation of MWCNTs did not appear to improve the sporeling release properties of PDMS alone.	[76]
		*Mytilus galloprovincialis* (mussels, pediveligers, and plantigrades)	*In vitro* study Static assay Filtered seawater 18 °C 48 h for pediveligers; 6 h for plantigrades	The incorporation of MWCNTs did not affect mussels’ adhesion or settlement, in comparison to PDMS alone.	[77]
		Bacteria and diatoms	*In situ* study (0.5–1.0 m below the surface) Natural seawater (Zhoushan, China) 28 days	The incorporation of MWCNTs altered the biomass and community composition of biofilms and subsequently decreased mussel settlement (up to around 20% settlement reduction) in comparison to bare PDMS.	[78]
		*Mytilus coruscus* (mussels, plantigrades)	*In vitro* study Autoclaved filtered seawater 18 °C, 12 h		
	Chlorinated rubber	Pioneer eukaryotic biofilm communities	*In situ* study (1.5 m below the surface) Natural seawater (Xiaoshi Island, China) 10 °C, 312 days	The incorporation of MWCNTs significantly improved AF effects by reducing the diversity and the abundance of pioneer eukaryotic microbes (significantly reduced mean species richness, *p* < 0.01).	[79]
Hydroxyl-modified MWCNTs	Silicone oil-infused PDMS	Marine bacteria	*In vitro* study Fresh seawater 28 °C, 10 days	Antiadhesion (up to 35% higher removal rate) and AF properties were enhanced, particularly when higher volume ratios of hydroxylated MWCNTs were used.	[80]
		Marine micro- and macrofoulers	*In situ* study (1–2 m below the surface) Natural seawater (Yellow Sea, China) 8 months		
Carboxyl and hydroxyl-modified MWCNTs	PDMS	Pioneer eukaryotic biofilm communities (such as sea slime, algae sporelings, invertebrates)	*In situ* study (0.8–1.5 m below the surface) Natural seawater (Xiaoshi Island, China) 2 months	The incorporation of MWCNTs showed excellent AF performance and effectively reduced colonization by pioneer eukaryotes, in comparison to plain PDMS (Shannon diversity index, *p* < 0.05).	[55]
		Marine micro and macrofoulers	*In situ* study (1.5 m below the surface) Natural seawater (Weihai Western Port, China) 11 °C, 56 days	The incorporation of a low amount of MWCNTs greatly improved the AF properties of PDMS coatings. However, most modified coatings demonstrated weak modulating effects on pioneer biofilm communities compared to plain PDMS.	[81]
Carboxyl and hydroxyl-modified MWCNTs	PDMS	Pioneer biofilm bacteria	*In situ* study (1.5 m below the surface) Natural seawater (Xiaoshi Island, China) 10–17 °C, 24 days	All CNT/PDMS composites decreased Proteobacteria biofilm formation, but increased Cyanobacteria biofilm development.	[82]
Graphitized MWCNTs					
Carboxyl-modified SWCNTs					
Nanomagnetite–hydroxyl-modified MWCNTs	Silicone oil-infused PDMS	Marine bacteria	*In vitro* study Fresh seawater 28 °C, 24 h	The novel coating exhibited excellent antibiofilm adhesion performance with up to 98% removal rate, compared with PDMS (50% removal rate).	[83]
Fluorinated MWCNTs	PDMS		Pseudobarnacle adhesion test method	The incorporation of fluorinated MWCNTs improved the FR properties by reducing the pseudobarnacle adhesion strength by 67% compared to bare PDMS, and by 47% compared to pristine MCWNT/PDMS.	[84]
	Silicon	*Escherichia coli*	*In vitro* study Phosphate-buffered saline 37 °C, 6 h	The incorporation of fluorinated MWCNTs showed a decrease of about 98% on CFUs when compared with bare silicon surfaces.	[85]

Abbreviations: AF—antifouling; CNTs—carbon nanotubes; FR—fouling release; MWCNTs—multi-walled carbon nanotubes; PDMS—polydimethylsiloxane; SWCNTs—single-walled carbon nanotubes.

**Table 3 antibiotics-11-01102-t003:** Adapted MINORS scale and mean score of the assessed studies.

Criterion	Mean Score
**1. A clearly stated aim:** the hypothesis or aim of the study is explicitly and precisely addressed.	2.00
**2. Adequate methodology:** the methods used to achieve the study’s aim are plausible and clearly described.	2.00
**3. Detection of bias:** data were collected according to an established protocol. At least 3 replicates or independent experiments were performed for each assay.	1.93
**4. Coating:** enough information is provided about the tested AF coating. 0: unclear 1: composition is indicated 2: composition AND production method are indicated	2.00
**5. Control group:** the study uses an appropriate control group, such as bare polymeric coating or uncoated surface.	1.93
**6. Surface characterization:** the study uses surface characterization methods to assess the coating’s properties.	1.87
**7. Type of study:** 1: *in vitro* study 2: *in situ* OR *in vitro* study under conditions representative of a real scenario	1.73
**8. Experimental setup:** sufficiently detailed description of the conditions under which the assays were performed, such as hydrodynamic conditions, culture medium, and temperature. 0: not described 1: incubation time OR culture medium OR temperature is indicated (*) 2: incubation time AND culture medium AND temperature are indicated (*)	1.90
**9. Organisms studied:** marine fouling organisms used to evaluate the coatings’ AF properties are clearly identified and representative. 0: not described 1: organism species OR organism quantity is indicated (*) 2: organism species AND organism quantity are indicated (*)	1.76
**10. Biofilm formation/fouling assay duration:** 0: not reported 1: short-term assay (≤48 h) 2: mid/long-term assay (>48 h)	1.76
**11. Results clarity:** the results of the study are presented in a clear and structured manner. 0: results are not clear 1: results are clear and easy to understand 2: results are clear and easy to understand AND quantitative results (e.g., cell concentration values and adhesion reduction percentage) are reported	1.83
**12. Statistical analysis:** the study includes the implementation and description of statistical tests appropriate to the dataset.	0.87

(*)—only applicable to *in vitro* studies; AF—antifouling.

## Data Availability

Not applicable.

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
