# Peer review of "Antifouling Performance of Carbon-Based Coatings for Marine Applications: A Systematic Review"

_antibiotics, 2022, doi:10.3390/antibiotics11081102_

Round 1

Reviewer 1 Report

The authors carried out a thorough and scientifically based analysis of the literature data on coatings based on graphene oxide and carbon nanotubes that prevent marine biofouling. However, during the review, some remarks emerged:

  1. In the introduction, it is necessary to expand the citations and increase the list of references to at least 80-90 articles

2. If possible, obtain permission to publish and add to the chapter the results and discussions of several illustrations on the effectiveness of the coatings you describe.

Author Response

Reviewer 1

The authors carried out a thorough and scientifically based analysis of the literature data on coatings based on graphene oxide and carbon nanotubes that prevent marine biofouling. However, during the review, some remarks emerged:

  1. In the introduction, it is necessary to expand the citations and increase the list of references to at least 80-90 articles

The authors inserted 38 new references in the Introduction section as suggested by the Reviewer.

  1. If possible, obtain permission to publish and add to the chapter the results and discussions of several illustrations on the effectiveness of the coatings you describe.

The authors provided three new figures in the Results and Discussion section, illustrating the effectiveness of some described coatings (lines 236, 256 and 263, respectively).

Reviewer 2 Report

A PRISMA-oriented systematic review was conducted to evaluate the antifouling performance of carbon-based coatings for marine applications. It concluded that due to the high self-cleaning and enhanced antimicrobial properties, as well as durability, these functionalized composites showed outstanding results in protecting submerged surfaces from the settlement of fouling organisms in marine settings. It’s valuable review for the prevention of biofouling in marine. The authors should consider the following issues to further improve the paper:

1. The symbols in formulas should be defined clearly to be understandable.

2. Explanation of the past researches is unclear. Besides, the biofouling development on the solid surfaces in the marine shares some similar properties with that biofouling occurring in membrane separation processes. Authors should pay some attention on this field and thus deepen their understanding of the knowledge of biofouling. Some references should be included: “Transparent exopolymer particles (TEPs)-associated protobiofilm: A neglected contributor to biofouling during membrane filtration”, “Electro-assisted CNTs/ceramic flat sheet ultrafiltration membrane for enhanced antifouling and separation performance”

3. The conclusion should be improved. Primary findings with quantitative evaluations are needed.

4. Keywords should be chosen better.

5. The manuscript needs to be reviewed (read) properly for minor grammatical errors.

6. What’s the most important, “conclusion” is the most important part of the paper. Considering its importance, authors have to show their conclusion of the study. Here, only main results and conclusion of them are expected. Authors should read some conclusions of others papers in order to make it clearer. Based on all appointments done, I see that the theme studied is very interesting, but the paper needs reformulation in order to be clearer and to present a scientific format.

In conclusion I think this paper needs a complete revision before it can be accepted.

Author Response

Reviewer 2

A PRISMA-oriented systematic review was conducted to evaluate the antifouling performance of carbon-based coatings for marine applications. It concluded that due to the high self-cleaning and enhanced antimicrobial properties, as well as durability, these functionalized composites showed outstanding results in protecting submerged surfaces from the settlement of fouling organisms in marine settings. It’s valuable review for the prevention of biofouling in marine. The authors should consider the following issues to further improve the paper:

The authors acknowledge the Reviewer's comment and addressed his/her suggestions to improve the manuscript (see below).

  1. The symbols in formulas should be defined clearly to be understandable.

The authors were unable to identify the formulas that the Reviewer was referring to, thus no corrections were implemented based on this suggestion.

  1. Explanation of the past researches is unclear. Besides, the biofouling development on the solid surfaces in the marine shares some similar properties with that biofouling occurring in membrane separation processes. Authors should pay some attention on this field and thus deepen their understanding of the knowledge of biofouling. Some references should be included: “Transparent exopolymer particles (TEPs)-associated protobiofilm: A neglected contributor to biofouling during membrane filtration”, “Electro-assisted CNTs/ceramic flat sheet ultrafiltration membrane for enhanced antifouling and separation performance”

 The authors provided additional information concerning the past research on AF coatings development (lines 76 - 97).

“Several approaches have been used to tackle biofouling. These strategies can prevent, and delay biofilm formation or control already developed biofilms [24,30,31]. While preventive measures comprise both antimicrobial and AF surfaces, control procedures can involve the use of bacteriophages, QS inhibitors, disruption of biofilm matrix, as well as the use of matrices with self-cleaning properties [32–36].

During the twentieth century, the development of AF coatings emerged as a new approach to protecting ship hulls from biofouling. Among all the coatings developed and tested, tributyltin (TBT) stands out as one of the most effective in biofouling mitigation [37]. However, TBT was exclusively used in the 1970s, in combination with several copper compounds, since it was discovered that TBT-based compounds were highly toxic for non-target organisms. Moreover, they were also able to persist in the marine environment for a longer period than initially estimated [38,39].

The prohibition of organotin compounds like TBT strongly motivated AF coating manufacturers to find additional solutions that allow a similar biofouling mitigation performance obtained with TBT-based compounds but without adverse environmental impacts. The current challenge is to develop environmentally friendly marine AF coatings that are effective toward a broad range of marine organisms with no negative impact on non-target organisms. Furthermore, leached coating compounds should have a high degradation rate in the marine environment, as well as a low bioaccumulation rate [38].

Modifications of surface charge potential, hydrophobicity, and surface topography can constitute promising strategies [40]. In this context, the use of carbon nanomaterials as fillers for polymer composites is becoming increasingly relevant [41].”

Considering that biofouling on submerged marine surfaces shares some properties with biofouling occurring in membrane separation processes, the authors introduced the last concept and cited the references suggested by the Reviewer (lines 49 - 57).

“Biofouling is caused by the adhesion of microorganisms to surfaces, which rapidly grow and form a biofilm, building the basis for the further settlement of macroorganisms [13,14]. This biological process has a high impact on several industries, compromising, for instance, the performance of membrane separation processes and water filtration devices and deteriorating marine surfaces [15–18]. In cooling water circuits from power plants, biofouling leads to efficiency losses and accelerates deterioration due to biocorrosion by about 20% [19]. In turn, in marine environments, submerged surfaces are rapidly colo-nized by different types of micro- and macroorganisms.”

  1. The conclusion should be improved. Primary findings with quantitative evaluations are needed.

The conclusions of this manuscript were improved according to the Reviewer's comments (see the answer to point 6).

  1. Keywords should be chosen better.

The authors redefined the keywords as suggested by the Reviewer.

“Carbon nanomaterials; Graphene; Graphene oxide; Carbon nanotubes; Antifouling coatings; Antibiofilm activity; Marine biofouling.”

  1. The manuscript needs to be reviewed (read) properly for minor grammatical errors.

 The manuscript has been carefully revised in order to correct minor grammatical errors.

  1. What’s the most important, “conclusion” is the most important part of the paper. Considering its importance, authors have to show their conclusion of the study. Here, only main results and conclusion of them are expected. Authors should read some conclusions of others papers in order to make it clearer. Based on all appointments done, I see that the theme studied is very interesting, but the paper needs reformulation in order to be clearer and to present a scientific format.

The Conclusions section was improved according to the Reviewer´s comments.

 “Among the solutions found to control marine biofouling, the application of coatings with antifouling properties on submerged surfaces has been the most promising approach. Due to their outstanding properties, carbon nanomaterials have proven to be promising in the production of antimicrobial and antifouling surface coatings.

This systematic review demonstrated that, over the last few years, there has been an increasing interest in the synthesis and evaluation of carbon-based coatings, in particular those containing GP or CNTs, to prevent and control marine biofouling. Most of the reviewed studies investigated the efficacy of modified/functionalized GP or CNTs, which demonstrated improved AF performance compared to their pristine forms.

Although these studies provided promising results, most AF coatings were only evaluated in vitro and, therefore, in situ or case studies are missing to validate their real-world application. It is also noteworthy that 26.7% of studies only qualitatively evaluated the AF performance of developed coatings or did not disclose the extent of attachment reduction and/or foulers inactivation. Also, the degradation or bioaccumulation rates of carbon-based coatings in the marine environment, as well as their toxicity for non-target organisms were not addressed in the reviewed studies.

The development of more accurate and reliable in vitro test methods that are able to mimic the hydrodynamic conditions observed for each target application is of paramount importance in obtaining reliable data. This is crucial because in situ tests should not be performed with surfaces that release compounds for which the leaching and toxicity profiles have not been determined. With an increasing environmental conscience from the public and regulatory organizations, novel coatings must be non-toxic to non-target organisms, durable, and amenable to large-scale production in a sustainable way. This is a multidisciplinary endeavor requiring the involvement of different stakeholders who have to find a common language to address these issues. In a globalized world depending on maritime transportation, the development of more effective AF coatings is of extreme importance.

Overall, this review illustrated the antifouling potential of carbon-based surfaces for marine applications and emphasized the need for further research on this topic.”

In conclusion I think this paper needs a complete revision before it can be accepted.

The authors implemented all recommendations to improve the manuscript.
